# Staff-Facilitated Telemedicine Care Delivery for Treatment of Hepatitis C Infection among People Who Inject Drugs

**DOI:** 10.3390/healthcare12070715

**Published:** 2024-03-25

**Authors:** Rebecca G. Kim, Claire McDonell, Jeff McKinney, Lisa Catalli, Jennifer C. Price, Meghan D. Morris

**Affiliations:** 1Department of Internal Medicine, Division of Gastroenterology and Hepatology, University of Utah, Salt Lake City, UT 84132, USA; rebecca.g.kim@hsc.utah.edu; 2Department of Epidemiology and Biostatistics, University of California San Francisco, San Francisco, CA 94158, USA; claire.mcdonell@ucsf.edu; 3Department of Medicine, Division of Gastroenterology and Hepatology, University of California San Francisco, San Francisco, CA 94143, USA; jeff.mckinney@ucsf.edu (J.M.); lisa.catalli@ucsf.edu (L.C.); jennifer.price@ucsf.edu (J.C.P.); 4Liver Center, University of California San Francisco, San Francisco, CA 94143, USA

**Keywords:** HCV treatment, telehealth, community-based care, marginalized populations

## Abstract

Background: Telemedicine offers the opportunity to provide clinical services remotely, thereby bridging geographic distances for people engaged in the medical system. Following the COVID-19 pandemic, the widespread adoption of telemedicine in clinical practices has persisted, highlighting its continued relevance for post-pandemic healthcare. Little is known about telemedicine use among people from socially marginalized groups. Methods: The No One Waits (NOW) Study is a single-arm clinical trial measuring the acceptability, feasibility, and safety of an urban point-of-diagnosis hepatitis C (HCV) treatment initiation model delivered in a non-clinical community setting. Participants enrolled in the NOW Study are recruited via street outreach targeting people experiencing homelessness and injecting drugs. Throughout the NOW Study, clinical care is delivered through a novel staff-facilitated telemedicine model that not only addresses geographic and transportation barriers, but also technology and medical mistrust, barriers often unique to this population. While clinicians provide high-quality specialty practice-based care via telemedicine, on-site staff provide technical support, aid in communication and rapport, and review the clinicians’ instructions and next steps with participants following the visits. Research questionnaires collect information on participants’ experience with and perceptions of telemedicine (a) prior to treatment initiation and (b) at treatment completion. Discussion: For people from socially marginalized groups with HCV infection, creative person-centered care approaches are necessary to diagnose, treat, and cure HCV. Although non-clinical, community-based staff-facilitated telemedicine requires additional resources compared to standard-of-care telemedicine, it could expand the reach and offer a valuable entrance into technology-delivered care for socially marginalized groups. Trial registration: NCT03987503.

## 1. Introduction

### 1.1. Background and Rationale

Hepatitis C virus (HCV) infection is a leading cause of chronic liver disease, affecting over 56 million people worldwide [1,2]. New HCV infections are estimated to be around 1.5 million per year through 2030 [2] with only a fraction of infections diagnosed and even fewer being treated [1,2]. Socially marginalized populations, including people experiencing homelessness and people who inject drugs (PWID), are disproportionately affected by HCV, with a rising incidence related to the opioid epidemic [3,4,5]. Remarkable progress has been made in HCV treatment, making HCV elimination theoretically possible despite the rise in injection drug use; with the use of direct-acting antiviral (DAA) therapies, sustained virologic response (SVR) or cure rates approach 100% [1,6]. To successfully connect PWID to HCV testing and treatment, innovative person-centered care approaches are required [7,8,9].

During the COVID-19 pandemic, healthcare delivery shifted due to shelter-in-place orders, and, as a result, telemedicine use expanded dramatically [10,11]. Even after the pandemic, with the extension of telehealth policies [12], telemedicine remains an integral part of healthcare delivery to overcome geographic barriers to medical care [10,12,13]. Telemedicine, when provided in community spaces, can expand healthcare access for people experiencing social marginalization including homelessness [14,15,16]. Telemedicine, in addition to other electronic health technologies, have been successfully implemented in the HCV care cascade, specifically to gain access to marginalized populations like PWID [17,18,19,20]. In one study, where telemedicine visits for HCV treatment were incorporated into opioid treatment programs (OTPs), participants reported high satisfaction with telemedicine that increased over the course of the study period [20]. In a systematic review, among studies investigating the use of telemedicine for HCV treatment in primary care and prisons, satisfaction with telemedicine among participants was also high [17]. These studies contribute important findings and provide evidence for the use of telemedicine for HCV treatment among marginalized groups. 

People with unstable housing or PWID face unique barriers to telemedicine [21]. These barriers may include a lack of access to necessary devices or reliable internet, limited prior computer experience or technical skills, mistrust of technology and healthcare organizations, and may be negatively impacted by healthcare provider bias [22,23]. Although prior studies describe interventions that have overall improved HCV care, our staff-facilitated telemedicine intervention hopes to overcome each of these common barriers for marginalized populations. 

### 1.2. Objectives

As a sub-study of the NOW Study clinical trial (NCT03987503), we aimed to assess the acceptability and feasibility of staff-assisted telemedicine delivered care for the community-based treatment of HCV among PWID. The primary aim is to assess participant experience and satisfaction with staff-facilitated telemedicine for HCV clinical care. Secondary aims include assessing the (a) implementation of the telemedicine intervention including feasibility based on participant utilization and (b) the impact of the intervention on HCV treatment adherence and SVR12 (cure). This novel care delivery approach addresses an urgent unmet need in the elimination of HCV as a public health threat [24] among a medically-underserved population with unique barriers to healthcare and telemedicine. 

## 2. Materials and Methods

### 2.1. Study Design 

The No One Waits (NOW) Study is a nonrandomized, single-arm clinical trial measuring the acceptability, feasibility, and efficacy of an urban community-based point-of-diagnosis HCV treatment initiation model (NCT03987503). A complete description of the NOW Study is published elsewhere [25] Briefly, potential participants are screened for HCV infection, and, if positive, offered HCV treatment with study-provided sofosbuvir/ledipasvir at the time of diagnosis. Prior to treatment initiation, but on the same day, participants complete an initial clinician visit via staff-facilitated telemedicine and a pre-treatment blood draw. Staff-facilitated telemedicine visits are then offered every 4 weeks through the 12 weeks of treatment and at 12 weeks post treatment (SVR12). Participants also complete telemedicine-specific questionnaires at their initial visit prior to treatment and at the time of treatment completion (Figure 1). 

Staff-facilitated telemedicine replaces standard healthcare delivery approaches for HCV treatment, including in-person clinic visits and telemedicine visits from people’s personal devices or computers. This model of care uses a consistent community research site conveniently located near areas where participants spend most of their time. The site provides space with access to a computer and reliable wireless internet. Additionally, trained staff members facilitate each telemedicine visit. Lastly, blood tests are collected on-site, eliminating the need to travel to and visit a laboratory separately (Figure 1). 

### 2.2. Setting

The study is conducted at a fixed non-clinical community space located in an urban neighborhood where PWID are known to congregate near public transit and social service organizations. Within the fixed community space, there are private spaces for research and clinical activities, including a dedicated space to conduct staff-facilitated telemedicine visits. During study activities, food, harm reduction supplies, and COVID-19 information and vaccination are available (Figure 1). 

### 2.3. Population to Be Studied and Eligibility Criteria

The study population includes participants among socially marginalized populations, specifically persons who inject drugs (PWID) and unstably housed people, who tested positive for HCV and were eligible for simplified HCV treatment. 

Eligible participants are 18 years or older, who report either injecting drugs in their lifetime or having a blood transfusion in or before 1992, confirmed to have HCV viremia and willing to receive simplified HCV treatment and participate in staff-facilitated telemedicine visits. Additional eligibility criteria were previously described [25]. 

### 2.4. Recruitment and Consent

Eligible participants are identified through methods previously described [25]. Participants provided written consent prior to the initiation of HCV treatment. 

Due to the challenges with the retention of participants from socially marginalized populations for research studies, participants’ contact information is reviewed at each visit, they are reminded of their following appointment via their preferred contact method 2–3 days prior, and additional contact is made through secondary/emergency contacts (if provided and authorized) for anyone who missed appointments. Participants also receive at least USD 20 at each return visit (totaling USD 260).

### 2.5. HCV Telemedicine Intervention

Once participants are diagnosed with HCV and prior to treatment initiation, they have their first staff-facilitated clinic visit. This visit includes a focused history and physical, medication review, and DAA counseling. Three additional visits are then conducted about every 4 weeks and include an assessment of treatment adherence and barriers to adherence, medication side effects, answering questions, and outlining the next steps. Following the 12 weeks of treatment, a final staff-facilitated telemedicine visit is conducted to assess for cure and review measures to prevent reinfection (Figure 2). 

### 2.6. Staff Members’ Training

In addition to standardized research training, each staff member completes educational training titled “Hepatitis C 101”. They learn the basics of liver physiology, viral hepatitides with a focus on HCV including how infection is spread, prevalence, features of chronic disease, challenges in cure, available tests for HCV infection, and important information regarding antibodies vs. virus, infection clearance, and re-infection. 

### 2.7. Staff Members’ Roles

#### 2.7.1. Set up Visit Space

Prior to participants arriving, for each visit, a staff member sets up the space in the community center including tables and chairs. To ensure adequate privacy during visits, dividers and a white noise machine are located in the visit space. The staff member also confirms that the clinician is available for the visit and has any relevant laboratory testing results to review (Figure 3). 

#### 2.7.2. Build Rapport

Once the participant arrives, staff members establish rapport and inquire about their HCV treatment. They make note of any concerns to ensure these are discussed with the clinician during the visit. The staff member then provides an overview of the telemedicine visit, describing steps and expectations. Before starting the visit, the staff member askes the participant if they would prefer to have privacy with the clinician and offers to leave the room during the visit. 

#### 2.7.3. Troubleshoot Technology

Staff members set up the telemedicine visit, which includes starting up the computer, connecting to wireless internet, joining the clinician’s zoom waiting room with audio and video enabled, and troubleshooting any technology issues that occur.

#### 2.7.4. Support Communication between Provider and Participant

Throughout the visit, the staff member aids communication by clarifying questions, repeating statements if there were any audio issues, and bringing up concerns the participant mentioned when they initially arrived that they may have forgotten during the visit. 

#### 2.7.5. Debrief with Participant

Following each visit, the staff member casually debriefs with the participant, then reviews the next steps outlined by the clinician and writes out a visit reminder slip that includes a list of next steps and the time of the participant’s next visit. 

### 2.8. Participants’ Experience

During their initial staff-facilitated telemedicine visit, participants complete a survey developed by our research team. The survey includes questions asking about (1) access to devices, (2) access to and use of various internet sources, (3) prior experience with telemedicine visits, (4) timing of telemedicine visits in relation to the COVID-19 pandemic, (5) details of their telemedicine use, including ease of use, technical assistance, scheduling, rapport with the clinician, and interest in future telemedicine use, and (6) experience with and thoughts about staff-facilitated telemedicine use during the NOW Study (Appendix A. Participants then complete their research and staff-facilitated telemedicine visits as previously described. At their treatment completion visit, they are asked additional survey questions regarding their experience and acceptance of the staff-facilitated telemedicine visits.

### 2.9. Clinician’s Approach

During the staff-facilitated telemedicine visits, the clinician provides a 20 min standard-of-care visit for anyone undergoing HCV treatment. At the initial visit, the hepatology provider asks about risk factors for HCV, symptoms related to liver disease, describes treatment, potential side effects, the importance of adherence, the laboratory monitoring required, and follow-up visits. At each subsequent visit, the provider will ask about medication adherence, any challenges receiving, keeping, or taking HCV medication, possible side effects observed, answer any questions, and review available laboratory data. If applicable, additional counseling regarding strategies to prevent misplacement and/or theft of HCV medication and harm reduction practices (e.g., not sharing needles) is provided if the participant is unstably housed or is a PWID. 

During the research study hours, the clinician is available for scheduled telemedicine visits. This flexibility also enables them to accommodate patient drop-ins, quickly address questions, and reschedule any missed appointments. This approach aims to enhance medication adherence and foster a stronger clinician–patient relationship.

### 2.10. Objectives of Clinic Visits

Telemedicine Visit 1: for participants with HCV infection, they undergo consent to enroll in the study and start HCV treatment. Their telemedicine visit includes a focused history and physical, medication review, and DAA counseling. Laboratory tests are collected on-site, they receive 2 weeks of medication, and are scheduled for a follow-up research visit in 2 weeks. 

Follow up visits: participants receiving HCV treatment are scheduled for follow-up visits every 2 weeks—research and staff-facilitated telemedicine visits alternated every 2 weeks through completion of treatment. They are then scheduled for post-treatment visits that include a research visit and survey completion at 4 weeks post-treatment and a final staff-facilitated telemedicine visit at 8 weeks post-treatment, along with repeat HCV laboratory testing for confirmation of cure. 

End of care: cured participants are educated on how they can decrease their risk of reinfection and improve their overall health. Participants who are not cured but remain interested in ongoing care are provided with the location and hours of a local mobile HCV treatment site and a hepatology clinic. 

### 2.11. Data Management and Analysis

We plan to assess the feasibility and acceptability of our staff-facilitated telemedicine visit. Feasibility will be determined by the number of visits completed, specifically the proportion of participants who completed 1 through 4 of their scheduled visits and the proportion who attended additional drop-in visits. Acceptability will be measured based on survey responses regarding participants’ attitudes toward telemedicine use, their experience with telemedicine vs. staff-assisted telemedicine, and participants’ interests in future telemedicine visits compared to other healthcare delivery methods like in-person visits.

## 3. Discussion

This study aims to assess the acceptability and feasibility of staff-facilitated telemedicine to improve access to care for unstably housed and PWID with HCV infection living in an urban setting. Our intervention uses an innovative approach to overcome several barriers to telemedicine unique to this population. Transportation and geographic barriers are addressed by conducting telemedicine visits in a convenient, reliable, and consistent location within the community. A private space, dependable wireless internet, and an accessible computer are all provided by the research team. Lastly, possible mistrust of technology or healthcare is mitigated through the relationship formed with the staff member facilitating the telemedicine visit in-person. 

Prior studies have incorporated remote healthcare to successfully treat HCV infection among marginalized populations, specifically within prisons, primary care clinics, and OTPs [17,20]. Our intervention, while similar, brings telemedicine to non-clinical settings located within the community, separate from opioid treatment and other healthcare. More recently, a similar, community-based protocol was proposed to use peer-facilitated telemedicine for HCV treatment among PWID in rural communities [26]. It also implements an innovative, personalized approach to HCV treatment among a marginalized population. Despite the similarities, each intervention is conducted in a unique population; while other studies assess what works in OTPs, prisons, and rural communities, our intervention will provide care to PWID who are unstably housed in an urban setting. Due to the unique barriers experienced by each of these populations, their response to interventions for HCV care may differ; therefore, each study is critical and informative. Overall, our proposed protocol contributes to the growing effort to eliminate HCV infection by connecting medically-underserved populations to healthcare that has historically been inaccessible.

For people from socially marginalized groups with HCV infection, creative person-centered approaches are necessary for successful care throughout the HCV treatment cascade. While this non-clinical, community-based staff-facilitated telemedicine requires additional resources compared to standard-of-care telemedicine, it may offer a valuable entrance into alternative healthcare delivery modalities for socially marginalized groups.

### 3.1. Strengths and Limitations of the Study

The NOW Study does have some limitations. The staff-facilitated telemedicine intervention requires resources that may not be readily available in other settings. Paid staff are required, as well as a safe space with reliable wireless internet to conduct visits located within the community and a computer. These required resources may make the intervention difficult to replicate or sustain, limiting its generalizability. Additionally, the study does not include a control group, so there is no direct comparison between staff-assisted telemedicine visits and standard-of-care telemedicine or in-person visits.

Strengths of the intervention include the convenient and consistent location easily accessible to our target population, the flexibility of visits to accommodate participants, and the role of the staff members to cultivate an environment of trust and familiarity, and to offer hands-on assistance for any technical challenges or difficulty with communication. With its use of staff-facilitated telemedicine visits, the NOW Study will assess the acceptability and satisfaction with remote care delivery among hard-to-reach populations. If feasible and acceptable, this community-based intervention can serve as a model for the expansion of HCV treatment to other marginalized populations.

### 3.2. Dissemination Plan

The study’s findings have been and will be disseminated in traditional ways including abstract submissions and poster presentations at national conferences and manuscripts published in peer-reviewed journals. Additionally, study results will be shared with the general public during a community night and via community handouts disseminated through peer navigators and community organizations.

## 4. Conclusions

Staff-facilitated telemedicine is an innovative person-centered approach to diagnose, treat, and cure HCV among people from marginalized groups. It requires more resources than standard-of-care telemedicine, however it has the potential to overcome unique barriers to HCV treatment that marginalized groups experience. Staff-facilitated, community-based telemedicine expands the reach of HCV care to people who use drugs, and helps to make HCV elimination possible. 

## Figures and Tables

**Figure 1 healthcare-12-00715-f001:**
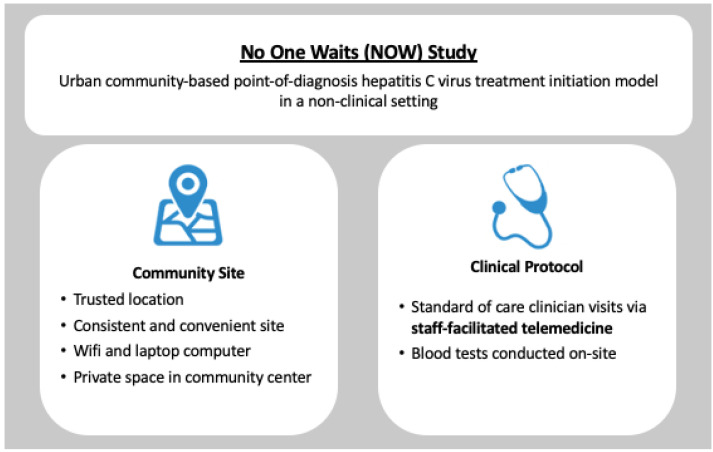
Staff-facilitated telemedicine model.

**Figure 2 healthcare-12-00715-f002:**
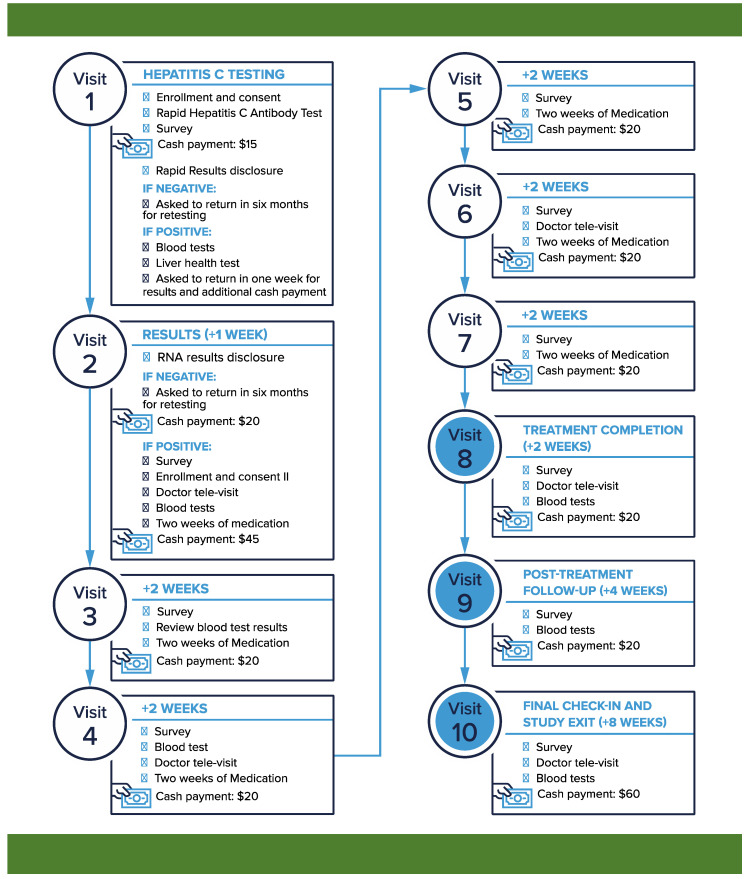
No One Waits Study visit schedule.

**Figure 3 healthcare-12-00715-f003:**
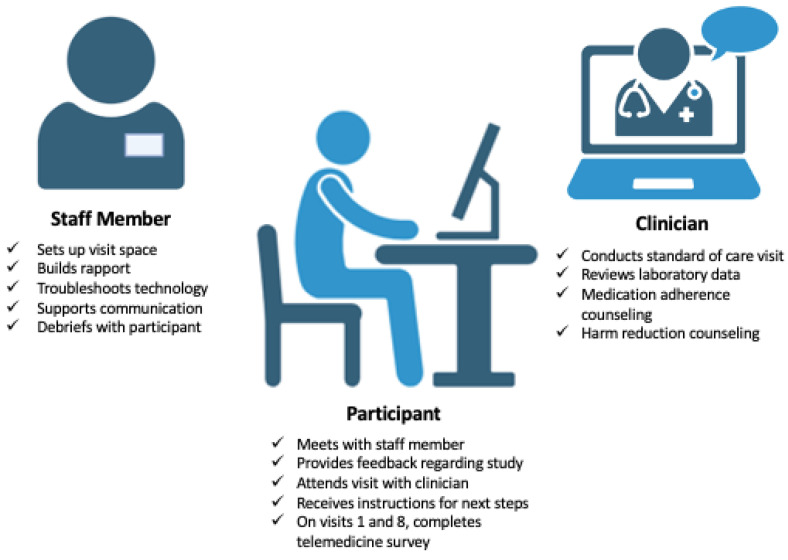
Roles of staff, participant, and clinician during staff-facilitated telemedicine visits.

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
