# Peer review of "Staff-Facilitated Telemedicine Care Delivery for Treatment of Hepatitis C Infection among People Who Inject Drugs"

_healthcare, 2024, doi:10.3390/healthcare12070715_

Round 1
Reviewer 1 Report
Comments and Suggestions for Authors
Dear author team,
with much interest did I read your study protocol. I am not a medical doctor, so I am not in the position to comment on the treatment regimen presented in the study protocol. The telehealth approach seems very interesting, specifically for the outlined target group. You are interested to make a scientific contribution on the following concepts: access to service delivery. In detail, you aim to assess participant experience and satisfaction with staff-facilitated telemedicine. Further you aim to assessing a) implementation of the telemedicine intervention including feasibility based on participant utilization and b) the impact of the intervention on HCV 75 treatment adherence and SVR12 (cure).
Overall, in my opinion the intervention/protocol is well elaborated, I just have four points I would like to get more information from the author team as reader and public health expert:
1. Reading the study protocol, I am not really convinced on the description of the survey used for participant experience and satisfaction with the staff facilitated approach. I suggest that the tools to be used need to be explained in more detail (self developed tool/scales? validated/tested tool? what are the items?, etc.).
2. Also, I would be interested in some elaboration on the sustainability of the tested approach/intervention...how could the suggested telehealth approach be embedded in a health systems strategy vs. a one off tested approach. What are you aiming to contribute to the field of digital approaches in public health/clinical interventions?
3. A more elaborated section on the included participants is needed, to be able to publish results of the intervention later: more details on the inclusion criteria, gender, age, etc. how many participants are actually targeted with the intervention?
4. Your intervention targets a highly vulnerable population group. I did not read much in your protocol about your do-no-harm approach. Suggesting to elaborate on issues of data protection, confidentiality, payment for adherence etc.
As said beforehand, I found your study/intervention approach very interesting and wishing all the best for the intervention/study especially for the benefit of your vulnerable target group and that we can learn from your applied approach for future public health interventions using digital/telehealth approaches.
Author Response
Reviewer 1:
Overall, in my opinion the intervention/protocol is well elaborated, I just have four points I would like to get more information from the author team as reader and public health expert:
- Reading the study protocol, I am not really convinced on the description of the survey used for participant experience and satisfaction with the staff facilitated approach. I suggest that the tools to be used need to be explained in more detail (self developed tool/scales? validated/tested tool? what are the items?, etc.).
Thank for this comment, and we agree additional information about our survey would be useful. It is a self-developed tool and we have included the survey questions as supplemental material with our revised manuscript.
- Also, I would be interested in some elaboration on the sustainability of the tested approach/intervention...how could the suggested telehealth approach be embedded in a health systems strategy vs. a one off tested approach. What are you aiming to contribute to the field of digital approaches in public health/clinical interventions?
This is an excellent question. The primary aims of the study are to determine the efficacy, acceptability, and safety of this intervention. Evaluating the feasibility of reproducing this model in other settings is an important second aim. Our overall goal is to provide a model that will be implemented more widely in community-based settings.
- A more elaborated section on the included participants is needed, to be able to publish results of the intervention later: more details on the inclusion criteria, gender, age, etc. how many participants are actually targeted with the intervention?
We describe in section 2.3 that eligible participants are 18 years or older, who report either injecting drugs in their lifetime or having a blood transfusion in or before 1992, confirmed to have HCV viremia and willing to receive simplified HCV treatment and participate in staff-facilitated telemedicine visits. Additional eligibility criteria were described in a prior publication, Morris MD, McDonell C, Luetkemeyer AF, et al., JAMA Network Open 2023.
- Your intervention targets a highly vulnerable population group. I did not read much in your protocol about your do-no-harm approach. Suggesting to elaborate on issues of data protection, confidentiality, payment for adherence etc.
Our do-no-harm approach is described in detail in a previously published manuscript, Morris MD, McDonell C, Luetkemeyer AF, et al., JAMA Network Open 2023
As said beforehand, I found your study/intervention approach very interesting and wishing all the best for the intervention/study especially for the benefit of your vulnerable target group and that we can learn from your applied approach for future public health interventions using digital/telehealth approaches.
We greatly appreciate Reviewer 1’s comments, questions, and feedback.
Reviewer 2 Report
Comments and Suggestions for Authors
The topic of the study is relevant because the use of telemedicine among people from socially marginalized groups is little studied. Also, chronic hepatitis C continues to be a socially significant disease and widespread, despite effective modern direct-acting antiviral agents. The authors of the manuscript have comprehensively presented the study protocol, primary and secondary endpoints.
- The main question that the study analyzed is how acceptable, feasible and safe is a model of urban point-of-diagnosis, non-clinical community-based hepatitis C (HCV) treatment initiation, that has been implemented in marginalized social groups.
- Several aspects of the study are original: the use of telemedicine in a specific population - homeless people who inject drugs; starting the treatment of hepatitis C in non-clinical settings and, last but not least, the possibility of including marginalized groups in the treatment, which would reflect on limiting the epidemic of hepatitis C.
- Telemedicine gained experience during the COVID pandemic. As the authors point out, there is insufficient data on the application of telemedicine in special populations such as those included in the study. There are also insufficient data on the initiation and follow-up of hepatitis C treatment in the non-clinical setting.
- What the study adds to the published data to date is that it offers a model for the effective use of telemedicine in homeless and intravenous drug users. The presented model can also be used for other socially significant diseases that are more prevalent in these marginalized populations. Using this model, methods can be developed to educate marginalized groups and use telemedicine to initiate treatment for other diseases, as well as their follow-up in non-clinical settings.
- The researchers used the simplest single-arm study design. Accordingly, the limitations of this type of design are also seen in the presented research. The idea of this study can be enriched and continued in a randomized study.
- The main goals have been achieved - namely the use of telemedicine to initiate and monitor hepatitis C treatment in a non-clinical setting in the specific population of people. There are insufficient data on treatment outcomes, baseline clinical characteristics of patients, and long-term outcomes of the treatment, which are the limitations of the study.
- The references are relevant to the research data and the necessary discussions. The authors may add more resources regarding the treatment of hepatitis C in marginalized groups.
- Authors may include examples of questionnaires used. I have not recommendations for the figures presented in the manuscript.
I would only recommend analyzing the database in the literature on the issue of telemedicine in general and completing the bibliography
Author Response
Reviewer 2:
The topic of the study is relevant because the use of telemedicine among people from socially marginalized groups is little studied. Also, chronic hepatitis C continues to be a socially significant disease and widespread, despite effective modern direct-acting antiviral agents. The authors of the manuscript have comprehensively presented the study protocol, primary and secondary endpoints.
- The main question that the study analyzed is how acceptable, feasible and safe is a model of urban point-of-diagnosis, non-clinical community-based hepatitis C (HCV) treatment initiation, that has been implemented in marginalized social groups.
We agree that this is the primary aim of the study. As noted in response to Reviewer 1, our hope is that this model can be scalable and implemented in other settings to improve access to HCV care.
- Several aspects of the study are original: the use of telemedicine in a specific population - homeless people who inject drugs; starting the treatment of hepatitis C in non-clinical settings and, last but not least, the possibility of including marginalized groups in the treatment, which would reflect on limiting the epidemic of hepatitis C.
Thank you for this comment, and we agree that Reviewer 2 has highlighted several aspects that make our intervention unique and impactful.
- Telemedicine gained experience during the COVID pandemic. As the authors point out, there is insufficient data on the application of telemedicine in special populations such as those included in the study. There are also insufficient data on the initiation and follow-up of hepatitis C treatment in the non-clinical setting.
We agree that our staff-facilitated telemedicine intervention and our participant survey will make a notable contribution to the paucity of data on telemedicine use among our study’s population and HCV treatment outside of the clinical setting.
- What the study adds to the published data to date is that it offers a model for the effective use of telemedicine in homeless and intravenous drug users. The presented model can also be used for other socially significant diseases that are more prevalent in these marginalized populations. Using this model, methods can be developed to educate marginalized groups and use telemedicine to initiate treatment for other diseases, as well as their follow-up in non-clinical settings.
We want to thank Reviewer 2 for these points, and we completely agree. Although this intervention is specifically for HCV treatment in our study, one goal is for this to be replicated to enhance the care of chronic diseases among populations with barriers to healthcare access.
- The researchers used the simplest single-arm study design. Accordingly, the limitations of this type of design are also seen in the presented research. The idea of this study can be enriched and continued in a randomized study.
We agree that our results may be limited by the absence of a control group. However from an HCV elimination and public health standpoint, our purpose was to diagnose and cure HCV infection in as many participants as possible using an enhanced telemedicine intervention.
- The main goals have been achieved - namely the use of telemedicine to initiate and monitor hepatitis C treatment in a non-clinical setting in the specific population of people. There are insufficient data on treatment outcomes, baseline clinical characteristics of patients, and long-term outcomes of the treatment, which are the limitations of the study.
Thank you for these points – we intend to include the clinical characteristics, treatment and long-term outcomes in our future manuscript describing the impact of the intervention.
- The references are relevant to the research data and the necessary discussions. The authors may add more resources regarding the treatment of hepatitis C in marginalized groups.
We have added additional references to our revised paper to include a more complete bibliography.
- Authors may include examples of questionnaires used. I have not recommendations for the figures presented in the manuscript.
As stated in our comments to Reviewer 1, we have included our survey questions as supplemental material with our revised manuscript.
I would only recommend analyzing the database in the literature on the issue of telemedicine in general and completing the bibliography.
We have added three additional studies previously published describing the experience of treating HCV in marginalized populations, and two additional studies on telemedicine.
Reviewer 3 Report
Comments and Suggestions for Authors
Interesting thought. However, we are in the post-COVID era. There were some problems in the study:
1. How about ultrasound or FIBROSCAN/ARFI...? The role of doctor in the study seemed weak maybe could be replaced by AI.
2.How is the efficacy (SVR rate? compliance? lost follow-up rate?) of the program? PWID group often had poor adherence.
Comments on the Quality of English LanguageNil
Author Response
Reviewer 3:
Interesting thought. However, we are in the post-COVID era.
We understand that many health care visits and services are now conducted in-person. However, the healthcare system learned that remote care via telemedicine is a viable option and has still been continued in the post-COVID era. We argue that remote care can be particularly useful for populations with geographic barriers and the inability to travel to or mistrust of the traditional clinical setting.
There were some problems in the study:
- How about ultrasound or FIBROSCAN/ARFI...? The role of doctor in the study seemed weak maybe could be replaced by AI.
Thank you for these points. A fibrosis assessment using FIB-4 is performed for each participant prior to initiation of HCV treatment.
The role of the doctor is to conduct a standard-of-care HCV treatment clinic appointment. We have never previously used AI to conduct these visits and we want to model the care delivered via telemedicine after any other in-person or remote clinic visit performed for other hepatology patients.
- How is the efficacy (SVR rate? compliance? lost follow-up rate?) of the program? PWID group often had poor adherence.
These measures of efficacy will be assessed and reported in our manuscript describing the outcomes of our staff-facilitated telemedicine intervention.